# Antimicrobial stewardship hindered by inadequate biosecurity and biosafety practices, and inappropriate antibiotics usage in poultry farms of Nepal–A pilot study

Ajit Poudel [1,2], Shreeya Sharma [1], Kavya Dhital[1], Shova Bhandari[2], Pragun Gopal Rajbhandari[2], Rajindra Napit [1,2], Dhiraj Puri[1], Dibesh B. Karmacharya [1,2,3]*

**1** One Health Division, Biovac Nepal, Nala, Nepal, **2** One Health Division, Center for Molecular Dynamics Nepal, Kathmandu, Nepal, **3** Department of Biological Sciences, University of Queensland, Brisbane, Australia

* dibesh@biovacnepal.com, dibesh@cmdn.org

**Data Availability Statement:** All relevant data are within the paper and its Supporting Information files.

## Abstract

Nepal's poultry industry has experienced remarkable growth in the last decade, but farm biosafety and biosecurity measures are often overlooked by farmers. As a result, farms often suffer from sporadic and regular outbreaks of many diseases, impacting production and creating public health challenges. Poor management practices, including overuse of antibiotics for prophylaxis and therapeutics, can enhance the spread of poultry diseases by propagating antimicrobial resistance (AMR) that is threatening poultry and human health. We assessed biosafety, biosecurity risks and AMR stewardship in sixteen poultry farms located in four districts: Ramechhap, Nuwakot, Sindhupalchowk, and Kavre. Risk assessment and AMR stewardship evaluation questionnaires were administered to formulate biosafety and biosecurity compliance matrix (BBCM). Risk assessment checklist assessed facility operations, personnel and standard operating procedures, water supply, cleaning and maintenance, rodent/pest control and record keeping. Oral and cloacal samples from the poultry were collected, pooled, and screened for eight poultry pathogens using Polymerase Chain Reaction (PCR) tests. Based on BBCM, we identified the highest BBCM score of 67% obtained by Sindhupalchowk farm 4 and the lowest of 12% by Kavre farm 3. Most of the farms (61.6%) followed general poultry farming practices, only half had clean and well-maintained farms. Lowest scores were obtained for personnel safety standard (42.4%) and rodent control (3.1%). At least one of the screened pathogens were detected in all farms. *Mycoplasma gallisepticum* was the most common pathogen detected in all but three farms, followed by *Mycoplasma synoviae*. More than half of the farmers considered AMR a threat, over 26% of them used antibiotics as a preventive measure and 81% did not consider withdrawal period for antibiotics prior to processing of their meat products. Additionally, antibiotics classified as "Watch" and "Restrict" by the WHO were frequently used by the farmers to treat bacterial infections in their farms.

**Funding:** This study was made possible by the funding support from the Regional Environment, Science, Technology and Health (ESTH) Office for South Asia- the US State Department (Grant No. SNP40021GR3038). The funders did not play any role in the study design, data collection, and analysis, decision to publish, or preparation of this manuscript.

**Competing interests:** The authors have declared no competing interests exist.

## Introduction

Nepal's poultry industry has seen a significant and rapid growth in the last decade, contributing more than 4% to the national gross domestic product (GDP) [1, 2]. Majority of the poultry products are supplied by numerous commercial farms (54% of total poultry production) scattered throughout the country. Backyard poultry also accounts for significant proportion of the total poultry production (46%); poultry meat and eggs are an easy source for protein and livelihood [1, 3]. Rapidly expanding commercial poultry is reared in 64 out of 77 districts of Nepal and has an annual growth rate of over 18% [1, 4]. According to the latest poultry census by the Nepal Central Bureau of Statistics [5], majority of chickens reared in Nepal are broilers (87%), with only small number of farms keeping layer chickens (11%). Almost half of the poultry production (46%) comes from the central region of the country. Chitwan, Kathmandu, and Kaski districts account for more than 85% of total meat and eggs production.

In Nepal, in spite of burgeoning poultry industry, proper biosecurity measures are often overlooked [6]. Backyard poultry farmers often feel the burden of maintaining biosecurity due to lack of knowledge and perceived additional cost [7]. Biosafety encompasses measures to prevent transmission of infectious diseases, and biosecurity measures are meant to prevent introduction and spread of pathogens in farms. By implementing a proper biological containment (and exclusion) along with traffic control, segregation, and sanitation- an effective biosafety and biosecurity can be maintained [8]. Keeping healthy flocks not only guarantees financial security for the farmers, it can also prevent outbreaks of zoonotic diseases such as highly pathogenic avian influenza (HPAI) [2, 9]. With increased commercial poultry production, maintaining biosecurity and biosafety measures in farms have been challenging [6]. Lack of government initiatives (and efforts), both in developing and developed countries alike, to raise awareness and implement regulations on biosecurity and biosafety have also grossly undermined proper safe farming practices [10]. Although Veterinary Standards and Drug Administration Office (VSDAO) in Nepal has developed a manual for proper poultry management including biosecurity guidelines, it isn't properly enforced and is often overlooked by farmers [11]. Poultry productions that are primarily focused on profitability, with compromised biosafety and biosecurity practices for cost saving, will eventually face production loss and increased health risks to both birds as well as humans (and other animals) [8].

Despite increasing occurrence of disease outbreaks such as Avian Influenza (AI) in poultry farms, many farmers in Nepal are unaware of the importance of biosecurity measures and their practices [7]. This lack of awareness, along with inadequate enforcement of biosecurity regulations, is exacerbating the risk of AI transmission in the country [12]. Poor farm management practices resulting in AI infection in poultry and use of their unprocessed poultry waste in vegetable cultivation can contribute to the spread of AI [13]. Lack of information available to farmers regarding safe poultry management or diseases is not an issue in Nepal. The government uses variety of media outlets to disseminate information pertaining to public health and pandemic preparedness guidelines and additionally, farmers and entire communities share their information and knowledge among each other regarding diseases and their transmission dynamics [14]. As aforementioned, a guideline on poultry biosafety and biosecurity has also been developed by the Veterinary Standards and Drug Administration Office (VSDAO) [11]. The issue seems to be either lack of knowledge or disregard of preventative measures that can be implemented to prevent poultry diseases. A knowledge, attitude, and practice study conducted in Nepal in 100 poultry farms stated although the farmers were aware of the contagious properties of AI, only half of them thought that it could be prevented [14]. Similar trend was observed in another study that analyzed commercial poultry in 10 districts of Nepal, in which, only 10% of the farms had comprehensive biosafety protection for the farmers. They were

aware of the need for biosafety and biosecurity measures on a farm and had some form of personal protective equipment (PPE) and disinfecting agents but were inadequate [15].

Disease outbreaks in poultry farms due to bacterial pathogens such as *Mycoplasma gallisepticum* (Mg), *Mycoplasma synoviae* (Ms), *Escherichia Coli* (*E. coli*) and *Salmonella* also occur due to lapses in biosafety and biosecurity. Rampant and haphazard use of antibiotics in poultry farms in Nepal is a leading cause of antimicrobial resistance (AMR) and a looming threat to human health [16]. There is a need to reduce and promote responsible use of antibiotics in the poultry industry in Nepal [17, 18]. Antimicrobial stewardship, which is a coordinated program that promotes the appropriate use of antimicrobials (including antibiotics), reduces microbial resistance, and decreases the spread of infections caused by multidrug-resistant organisms, has become one of the important aspects of a comprehensive biosecurity and biosafety practices [7].

Kathmandu is a densely-populated metropolitan capital city of Nepal with a population of over 2 million people [19]. Any emerging, re-emerging and diseases of human health concern originating from animal production sites, such as poultry farms, can rapidly spread in a city like Kathmandu. Since disease outbreak, transmission and spread dynamics are directly linked to biosafety and biosecurity status of farms, it is vital to understand the current status of the farms located close to the city. We conducted a comprehensive risk assessment and status evaluation of biosafety, biosecurity and AMR stewardship in sixteen poultry farms located in four districts with high poultry production (Ramechhap, Nuwakot, Sindhupalchowk, and Kavre) surrounding the Kathmandu valley. In this study we aimed to better understand the implementation of biosafety and biosecurity measures in the selected poultry farms, the farmers' knowledge, attitude, and practice towards AMR, and the dynamics of diseases detected on their farms.

## Methodology

### Ethical approval

Our study followed ethical guidelines of the Department of Livestock Services (DLS) Nepal for the survey. Sampling and survey were conducted after obtaining written consent of farm owners. Biological samples were collected using proper biosecurity measures in the presence of the farm owner or caretaker. For human survey, ethical approval was obtained from Nepal Health Research Council (NHRC) (Reg. 411/2020).

### Study site and data collection

Four farms (small and medium sized; poultry <2000 per farm) in each of the four districts (Kavre, Sindhupalchowk, Ramechhap and Nuwakot) were selected for the study (Fig 1). To ensure uniformity among sampling sites within our available budget, four farms across the four districts were selected. Only commercial farms were selected as backyard poultry is more resilient to diseases [20] and they are not reared in large numbers as the commercial ones. Additionally, the sampling activities were carried out during the first year of Covid-19 pandemic (2020) and most farms were hesitant to consent to the study. Two broiler and two layer farms in each district were selected for this study (Table 1). All poultry samples were collected by trained veterinarian and veterinary technician. Ideally, 10% of total poultry would be sampled to obtain a representative data but as this was a pilot study, we utilized the practicality of Central Limit Theorem and collected samples of 30 chickens from each farm [21, 22], assuming normality of the sampling distribution and reduced variability to draw meaningful and reliable conclusions about the entire flock. Risk assessment checklist (S1 Table) and AMR stewardship questionnaire (S2 Table) were used to collect data. Risk assessment checklist

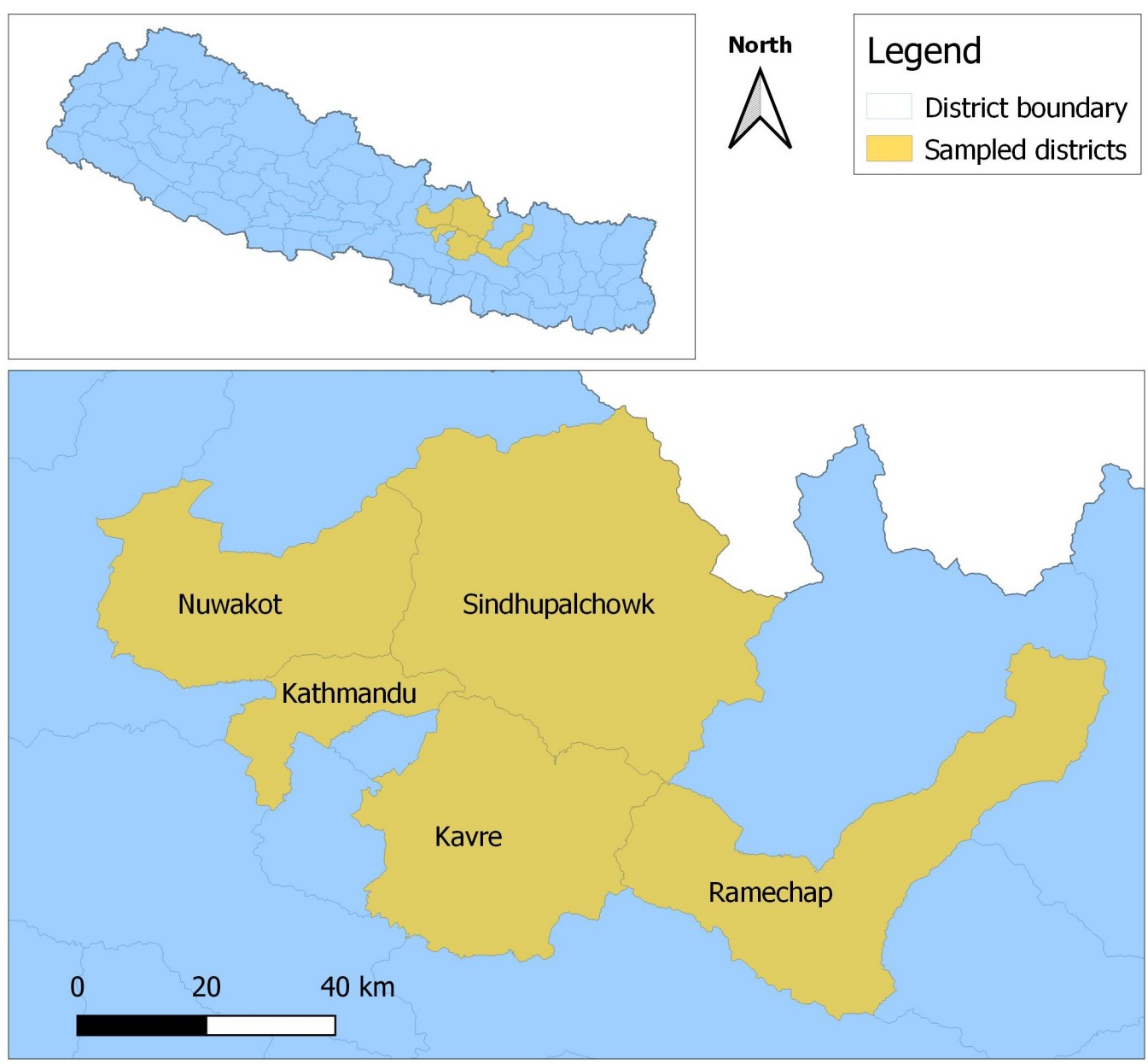

**Fig 1. Selected poultry farms in the districts (Kavre, Sindhupalchowk, Ramechhap and Nuwakot) surrounding Kathmandu valley.** The top map shows location of the districts in Nepal and the bottom map shows their location in respect with Kathmandu District (shown just for reference, not a sampling site). The map was generated using QGIS software, Version 3.30.0 [23]. The base map of Nepal administrative shape files was obtained from Open Data Nepal (http://opendatanepal.com).

assessed facility operations, personnel and standard operating procedures, water supply, cleaning and maintenance, rodent/pest control and farm record keeping.

### Biological sample collection

Oropharyngeal (n = 1) and cloacal (n = 1) swabs from each chicken were collected from selected farms (n = 16 farms; 30 birds per farm) and stored in viral transport media (VTM). All samples were stored in ice boxes (2−8˚C) during sample collection and transported in

**Table 1. Types, age, and total number of chickens in each farm.**

| Farms | Type of Chickens | Farm Size | Age of Chickens (Weeks) |
|---|---|---|---|
| Nuwakot Farm 1 | Broiler | 1200 | 5 |
| Nuwakot Farm 2 | Layer | 1000 | 38 |
| Nuwakot Farm 3 | Broiler | 800 | 3 |
| Nuwakot Farm 4 | Layer | 2000 | 55 |
| Kavre Farm 1 | Layer | 1500 | 62 |
| Kavre Farm 2 | Layer | 1100 | 28 |
| Kavre Farm 3 | Broiler | 600 | 4 |
| Kavre Farm 4 | Broiler | 500 | 7 |
| Ramechhap Farm 1 | Layer | 2000 | 20 |
| Ramechhap Farm 2 | Broiler | 1800 | 4 |
| Ramechhap Farm 3 | Broiler | 500 | 5 |
| Ramechhap Farm 4 | Layer | 1000 | 33 |
| Sindhupalchowk Farm 1 | Broiler | 900 | 7 |
| Sindhupalchowk Farm 2 | Layer | 2000 | 42 |
| Sindhupalchowk Farm 3 | Broiler | 1700 | 8 |
| Sindhupalchowk Farm 4 | Layer | 1500 | 50 |

liquid nitrogen container (-196˚C) to BIOVAC's lab in Kathmandu. The samples were then pooled (n = 10) from each farm (n = 6 pooled samples; pooled oropharyngeal swabs = 3 and pooled cloacal swabs = 3). These pooled samples were screened for eight poultry pathogens-Newcastle disease virus (NDV), Influenza A Virus (IAV), Infectious Bronchitis Virus (IBV), Infectious Bursal Disease (IBD), *Mycoplasma synoviae (Ms)*, *Mycoplasma gallisepticum (Mg)*, Marek's Disease Virus-1 (MDV1) and Marek's Disease Virus-2 (MDV2) using PCR.

## Nucleic acid extraction and PCR

The nucleic acid (DNA/RNA) from pooled samples were extracted using automated nucleic acid extractor (abGenix™ AITbiotech, Singapore) following manufacturer's instructions. The pooled swab samples were stored at -20˚C. PCR for detection of the eight pathogens were performed using SuperScript™ III Platinum™ One-Step qRT-PCR Kit w/ROX (Invitrogen, Catalog number 11745500). The primers for NDV, IBV, IBDV, Mg, Ms, MDV1 and MDV2 were designed using NCBI PrimerBlast® (Table 2).

For IAV, the primers IAV ISO_F and IAV ISO_R were used (Table 2) [15].

## PCR condition

PCR for each pathogen test was done in a 25ul reaction containing 4 µL of extracted RNA (for NDV, IAV, IBD, IBV) and 4 µL of extracted DNA (for MG, MS, MDV1 and MDV2), 1 µL each of respective 10pm forward and reverse primers, 12.5 µL of 2X Mastermix with ROX, 0.2 µL of SuperScript™ III Platinum™ enzyme and 6.3 µL of Nuclease free water. All eight PCR were carried out with enzyme activation at 45˚C for 15 minutes followed by one cycle of initial denaturation at 95˚C for 5 minutes. PCR for RNA viruses consisted of 45 cycles of denaturation at 95˚C for 30 seconds, annealing at 59˚C for 30 seconds and extension at 72˚C for 20 seconds. PCR for MS, MG, MDV1 and MDV2 consisted of 10 cycles of denaturation at 95˚C for 30 seconds, annealing at 63˚C for 40 seconds and extension at 72˚C for 20 seconds followed by 35 cycles of denaturation at 95˚C for 30 seconds, annealing at 60˚C for 40 seconds and extension at 72˚C for 20 seconds. The final extension for all eight PCR were carried out at

**Table 2. PCR primers for each of the pathogen designed using PrimerBlast® along with the gene they encode for, primer sequence, size, and their reference sequences (GenBank accession number or citations).**

| Pathogen | Gene | Primer Sequence (5'- 3') | PCR amplicon size | GenBank Accession Number / Citations |
|---|---|---|---|---|
| NDV | Fusion protein (F) Gene | CTCAATGTCACTATTGAYGTGG<br>CTGAGGAGARGCATTKGCTAT | 316 bp | [15] |
| IAV | Matrix Gene | CTTCTAACCGAGGTCGAAACG<br>GGTGACAGGATTGGTCTTGTC | 156 bp | [15] |
| IBV | Nucleoprotein (N) Gene | GGTGATGACAAGATGAWYGAGGA<br>CTCCTCATCTGAGGTYAATGC | 387 bp | MK618759, MN509587, HM245924, HQ848267, HQ850618, KC008600, JX840411, HQ014604, JQ977697, LC634083, MT665806, MK581202, MN548285, KP118891, MN128087, KM658226, DQ490209 |
| IBDV | VP2 gene | CCTGAACTAGCAAAGAACCTG<br>CAAGACGGTCCCTCTCACT | 97 bp | MN393076, MZ740264, MW316417, MK783981, EF517528, KU578102, GQ166970, MT935610, MN369418, OK043826, MN241438, MW483684, AM111353 |
| MS | 16S rRNA | CGTTCTCAGTTCGGATTGTAGTC<br>GTCGTCTCCGAAGTTAACAAACC | 170 bp | CP107525, CP107526, CP103982, CP069379, MN069582, LS991953, MH539126, MH539008, MF196168, KX259335 |
| MG | MGC2 Protein | GCTGGGTTGATTGTTGTTTCTT<br>TCTTCACGTTCTTGGATCATCAT | 95 bp | [24] |
| MDV1 | Glycoprotein B | AACATTAGACGACACCACAGCCATCTATAGCAGTGCAGCTC | 272 bp | MF431496, EU499381, KU744555, MG518371, DQ530348, KU744557, KT833852, JQ314003, AB049735, NC_002577, MH939248, AF282130, NC_002641, AF291866 |
| MDV2 | Glycoprotein B | TGACCGCCGTGTCTACTTGTCTCTTTCGTGTAGACCGACAG | 377 bp | AB049735, NC_002577, MH939248, AB024711, KU744557, KU744555, EU499381, U01886, MF431496, MG518371, DQ530348, KT833852, JQ314003, AF282130, NC_002641, AF291866 |

72˚C for 2 minutes. All the amplified PCR products were visualized using 1.5% Agarose Gel (S1–S5 Figs).

## Biosafety and biosecurity risk assessment

We created a Biosafety and Biosecurity Compliance Matrix (BBCM) score based on risk assessment checklist which included criteria such as facility operations, personnel and standard operating procedures, water supply, cleaning and maintenance, rodent/pest control and farm record keeping (S1 Table). General practices relate to infrastructure of or within the farm that aids in its biosecurity. Personnel standards and procedures include hygiene practices, use of disinfectants and precautionary measures taken to minimize pathogen contamination in farms. Water supply assessed whether the water given to poultry is adequately disinfected and clean. We also investigated preventive measures taken by farms to control rodents. Poultry rearing area and equipment cleanliness practices implemented by the farm personnel were also evaluated under cleaning and maintenance criteria. And finally, under record keeping criteria, we assessed practices of keeping records of daily activities, material usage/ consumption and any breaches detected in the farm.

For each selected criteria complied, one point was given for every activity implemented and a BBCM score was tallied for each category in every farm. A final farm BBCM score was then calculated and converted into percentage. We categorized farms that had >90% BBCM score as high, 60–89% as medium and <60% as low. Scores received by each farm are shown in Table 3 and visualized in Fig 2.

**Table 3. Biosafety and biosecurity risk assessment results.** Overall average BBCM score was calculated by dividing the sum of all BBCM fulfilled criteria (n) and calculating the percentage. The highest BBCM score (Sindhupalchowk Farm 4) was 67%, and the lowest BBCM score was 12% (Kavre Farm 3). The total number of activities assessed in each criterion are listed within parenthesis. Details of the activities assessed are shown in S1 Table.

| Location | General Practices (17) | | Personal Standards & Procedure (15) | | Water Supply (4) | | Rodent Control (4) | | Cleaning & Maintenance (10) | | Record Keeping (6) | | Overall Average | Diseases Detected |
|---|---|---|---|---|---|---|---|---|---|---|---|---|---|---|
| | n | % | n | % | n | % | n | % | n | % | n | % | % | |
| Nuwakot Farm 1 | 11 | 65 | 11 | 73 | 0 | 0 | 0 | 0 | 4 | 40 | 3 | 50 | 38 | Mg |
| Nuwakot Farm 2 | 12 | 71 | 6 | 40 | 3 | 75 | 0 | 0 | 5 | 50 | 3 | 50 | 48 | IBD |
| Nuwakot Farm 3 | 9 | 53 | 5 | 33 | 2 | 50 | 0 | 0 | 4 | 40 | 2 | 33 | 35 | IBD |
| Nuwakot Farm 4 | 14 | 82 | 8 | 53 | 2 | 50 | 0 | 0 | 5 | 50 | 2 | 33 | 45 | Mg |
| | | | | | | | | | | | **District Average (%)** | | **41.5** | |
| Kavre Farm 1 | 13 | 76 | 10 | 67 | 2 | 50 | 0 | 0 | 8 | 80 | 5 | 83 | 59 | Mg |
| Kavre Farm 2 | 14 | 82 | 6 | 40 | 3 | 75 | 0 | 0 | 7 | 70 | 2 | 33 | 50 | Mg |
| Kavre Farm 3 | 5 | 29 | 2 | 13 | 0 | 0 | 0 | 0 | 3 | 30 | 0 | 0 | **12** | Mg |
| Kavre Farm 4 | 6 | 35 | 3 | 20 | 2 | 50 | 0 | 0 | 4 | 40 | 0 | 0 | 24 | Mg |
| | | | | | | | | | | | **District Average (%)** | | **36.25** | |
| Ramechhap Farm 1 | 8 | 47 | 4 | 27 | 1 | 25 | 0 | 0 | 4 | 40 | 0 | 0 | 23 | Mg & Ms |
| Ramechhap Farm 2 | 13 | 76 | 3 | 20 | 3 | 75 | 1 | 25 | 6 | 60 | 3 | 50 | 51 | Mg & Ms |
| Ramechhap Farm 3 | 7 | 41 | 5 | 33 | 1 | 25 | 0 | 0 | 5 | 50 | 1 | 17 | 28 | Mg & Ms |
| Ramechhap Farm 4 | 10 | 59 | 4 | 27 | 1 | 25 | 0 | 0 | 4 | 40 | 0 | 0 | 25 | IAV & Mg |
| | | | | | | | | | | | **District Average (%)** | | **31.75** | |
| Sindhupalchowk Farm 1 | 9 | 53 | 6 | 40 | 1 | 25 | 0 | 0 | 4 | 40 | 5 | 83 | 40 | Mg |
| Sindhupalchowk Farm 2 | 13 | 76 | 8 | 53 | 3 | 75 | 0 | 0 | 8 | 80 | 5 | 83 | 61 | IAV |
| Sindhupalchowk Farm 3 | 10 | 59 | 10 | 67 | 1 | 25 | 0 | 0 | 3 | 30 | 5 | 83 | 44 | Mg |
| Sindhupalchowk Farm 4 | 14 | 82 | 11 | 73 | 3 | 75 | 1 | 25 | 6 | 60 | 5 | 83 | **67** | Mg |
| | | | | | | | | | | | **District Average (%)** | | **53** | |
| *Criteria Average (%)* | **61.6** | | **42.4** | | **43.8** | | **3.1** | | **50** | | **42.6** | | | |

Mg–*Mycoplasma gallisepticum*, Ms–*Mycoplasma synoviae*, IBD–Infectious Bursal Disease, IAV–Influenza A Virus

## Results

### Biosafety and biosecurity risk assessment

Analysis of overall Biosafety and Biosecurity of all farms surveyed showed low compliance (average BBCM score = <41%). The highest BBCM rating was scored by a farm in Sindhupalchowk (Farm 4, BBCM = 67%) and the lowest was by a farm in Kavre (Farm 3, BBCM = 12%). At district level, Sindhupalchowk had the most Biosafety and Biosecurity compliance (BBCM = 53%) whereas Ramechhap had the least (BBCM = 32%). Of all the assessed criteria, rodent control was the most neglected (BBCM = 3.1%). Only two farms (Ramechhap Farm 2 and Sindhupalchowk Farm 4) had implemented one out of four rodent control practices. General poultry farming practice was the most biosafety and biosecurity compliant criteria fulfilled by all farms (BBCM = 61.6%) (Table 3).

General practices, personnel standards and procedure, and cleaning and maintenance were the only categories implemented in all sixteen farms (Table 3, Fig 2). Only two farms (Ramechhap 2 and Sindhupalchowk 4) implemented rodent control, these were also the only farms that implement some activities of all six criteria. Water supply was implemented in all but two farms (Nuwakot 1 and Kavre 3). Similarly, record keeping was implemented in all but four farms (Kavre 3, Kavre 4, Ramechhap 1, and Ramechhap 4).

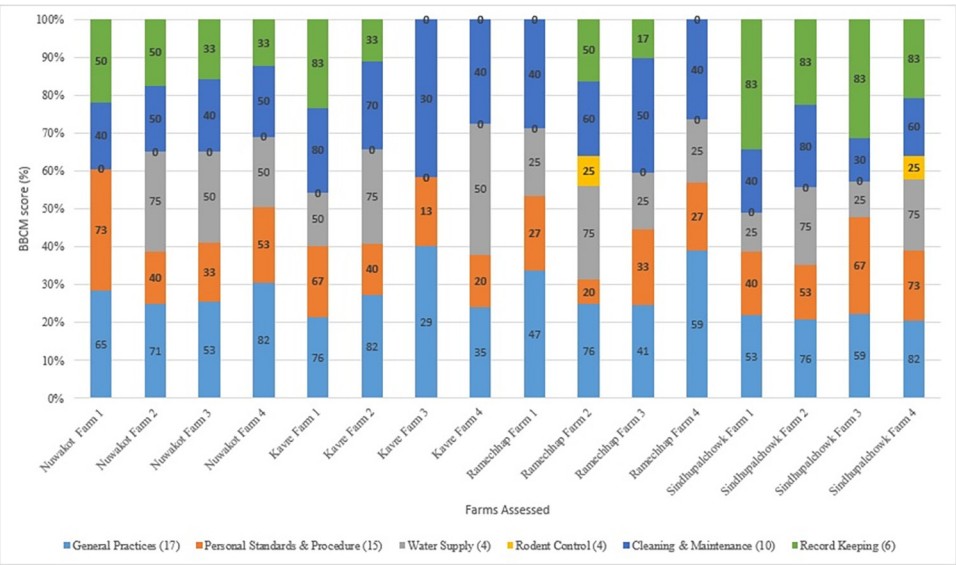

**Fig 2. Graphical representation of BBCM scores received by all 16 farms for various biosafety & biosecurity parameters.** Scores (percentage) received by all farms assessed in this study using BBCM score. Numbers in parenthesis in the legend refers to total number of activities assessed within each criterion.

Across all 16 farms, general practices received the highest criteria average (61.6%), followed by cleaning and maintenance (50%), water supply (43.8%), record keeping (42.6%), personnel standards and procedure (42.4%), and rodent control (3.1%).

At least one of the screened pathogens was detected in all farms. Mg was the most common disease detected, in all but one farm, followed by Ms. In Nuwakot, Mg [S1 Fig–(Mg)] and IBD [S1 Fig–(IBD)] were detected. IAV [S2 Fig–(IAV)] and both bacterial diseases, Mg [S4 Fig–(Mg)] and Ms [S4 Fig–(Ms)], were detected in Ramechhap. Only Mg was detected in both Sindhupalchowk [S3 Fig–(Mg)] and Kavre [S5 Fig (Mg)].

## AMR stewardship

A little over half (52%) of the surveyed farmers considered AMR as a real threat. Almost all farmers (93%) had not attended any programs or campaigns related to AMR. Very few farmers (12%) received training on AMR stewardship; these farmers received training/information on AMR from local animal health centres (21%), veterinarians (29%), vet technician (17%) or from vet suppliers (21%). Majority of the farmers (81%) did not consider implementing "withdrawal period" for antibiotics use prior to selling their meat products. Most of the famers trusted veterinarians (40%) and vet technician (14%) on receiving consultation on antibiotics use. Farmers used various antibiotics for prophylaxis (26%) and therapeutics (76%) needs (Table 4).

The farmers were also inquired about various types and proportion of antibiotics they used (Fig 3). Tetracyclines were the most used (35%) antibiotic class, followed by polymyxins (13%), quinolones (12%), aminoglycosides (12%), macrolides (8%), and beta-lactams (8%). Sulfonamide (7%) and pleuromutilin (5%) were the least used antibiotic classes.

## Discussion

Poor biosafety and biosecurity measures on poultry farms can lead to introduction and spread of bacterial and viral diseases. Regular monitoring and testing, an important component of

**Table 4. Knowledge and perception of AMR among poultry farmers.** Responses (in percentage) obtained from farmers on AMR-related questions.

| AMR Questions | | Percentage |
|---|---|---|
| Is AMR a threat? | Yes | 52 |
| | No | 48 |
| Where did you get AMR knowledge from? | Local Animal Health Center | 21 |
| | Veterinarian | 29 |
| | Vet Technician | 17 |
| | Vet Suppliers & Shops | 21 |
| | Self | 12 |
| Who do you most trust for consultation? | Veterinarians | 40 |
| | Vet Technician | 14 |
| | Self | 2 |
| | Local Animal Health Center | 26 |
| | Shops & Sales Team | 14 |
| | Local Animal Health Center | 2 |
| Do you consider withdrawal period after antibiotics use? | Yes | 12 |
| | No | 81 |
| | Don't Know | 7 |
| Withdrawal period prior to culling? | Yes | 14 |
| | No | 79 |
| | Don't Know | 7 |
| Any programs/campaigns attended on AMR? | Yes | 5 |
| | No | 93 |
| | Don't Know | 2 |
| Antibiotics used for prophylaxis? | Yes | 26 |
| | No | 74 |
| Antibiotics used for therapeutics? | Yes | 76 |
| | No | 24 |

implementing strict biosecurity and biosafety measures, are essential for detecting and containing infections in farms. Poultry farms with compromised biosafety and biosecurity compliance often suffer from various prevalent bacterial and viral infections such as Mycoplasma, IBD and IAV- affecting overall poultry health and lowering production.

Mg and Ms are bacterial diseases which spread through direct contact, respiratory secretions, or contaminated materials [25]. IBD, on the other hand, is a viral disease that can cause damage to the immune system of young chickens, often leading to mortality [26]. It is transmitted through contact with contaminated sources, such as faeces. IAV in poultry is primarily transmitted through direct contact with infected birds or through contact with contaminated surfaces, feed, or water. Wild birds, particularly waterfowl, are the natural reservoirs for the virus and are thought to be one of the major sources of avian influenza outbreaks [27, 28].

Introduction of infected birds, equipment, or materials in farms can result in the spread of diseases [29, 30]. Movement of people and animals on and off the farm can also contribute to the spread of disease [31]. Birds that are housed in crowded or unsanitary conditions are susceptible to diseases; maintaining proper biosecurity and biosafety measures in such farms are often challenging [32]. Implementation of these measures were severely lacking in the farms that we surveyed in our study. Only one farm scored medium biosafety and biosecurity compliance rating (Sindhupalchowk Farm 4, BBCM = 67%). None of the farms received a high score of >90%, and not surprisingly, diseases were detected in all farms (Table 3). Moreover,

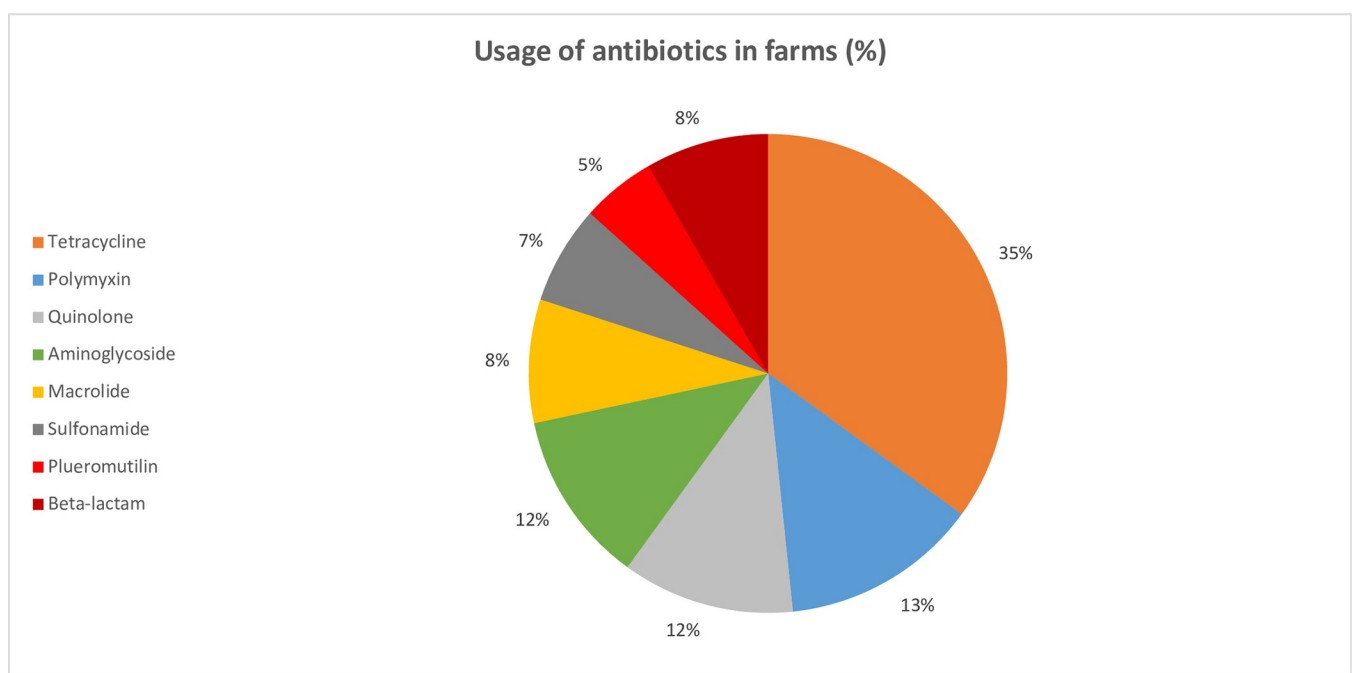

**Fig 3. Various types of antibiotics used (in %) in the surveyed poultry farms.**

out of all sixteen farms, only two (Ramechhap 2 and Sindhupalchowk 4) had evidence of conducting activities pertaining all six different categories of the assessment. Rest of the farms had activities/practices missing from an entire category.

Biosecurity measures, such as controlling the movement of people and animals onto the farm, implementing disinfection procedures, and providing protective clothing, can significant improve poultry production by reducing disease outbreaks and spread [33, 34]. We detected Mg and Ms in most of the farms, these bacterial infections have low mortality but associated morbidity directly affects egg and meat production [35]. Overall, the farms in Ramechhap district scored poorly in BBCM, something the district animal health authority needs to be aware of and make effort to improve.

Although majority of the farms focused on general practices, cleaning and maintenance, and personal standards and procedures, except for two farms (Ramechhap 2 and Sindhupalchowk 4) most had completely ignored rodent control (Table 3). Rodents are a major pest and a disease vector of poultry farms [36]. Rodents are also known to cause damage to farm infrastructure and feed, kill young chicks and eat eggs [37].

Poultry farms often resort to using medications to treat and salvage their flock only after disease outbreaks. We detected viral pathogens (IBD and IAV) and bacterial infections (Mg and Ms) in all but one farm (out of 16). Antibiotics was rampantly used, both as prophylaxis and therapeutics, by the farmers. Antibiotics families such as Tetracycline, Polymyxin, Quinolone, Aminoglycoside, and Macrolide were the most used by the farmers- consistent with national trend [38, 39]. More than a quarter (26%) of the farmers used antibiotics as prophylaxis as a preventive measure and 76% of farmers used them as therapeutics to treat diseases (Table 4). Detection of bacterial infections in the farms even after the use of all these antibiotics is deeply concerning.

Inappropriate (and haphazard) use of antibiotics in farms can develop AMR in bacteria, making them more difficult to treat. Such usage not only leads to altered composition and

diversity of gut microbiome in poultry [40, 41] but also result in high levels of resistance to several classes of antibiotics, including widely used antibiotics such as Polymyxin, Fluoroquinolones, and Beta-Lactams [42]. Further study needs to be carried out to assess presence of AMR genes in the bacterial pathogens detected in the farms.

The World Health Organization (WHO) has developed a classification system for antibiotics called Access, Watch, and Reserve (AWaRe) to guide the appropriate use of antibiotics and combat antimicrobial resistance. The Access group contains antibiotics that should be widely available and affordable, including first-line treatments for common infections. The Watch group contains antibiotics that should be used with caution and reserved for specific indications to prevent the development of resistance. The Reserve group contains last-resort antibiotics, which should be used sparingly and only when all other options have been exhausted [43]. Apart from tetracycline, the other two antibiotics that are classified as 'Access' by the WHO are used by farmers sparingly (Pleuromutilin– 7% and Penicillin– 5%). Antibiotics under 'Watch', such as Quinolone (12%), Aminoglycoside (12%) and Macrolide (9%) are widely used by poultry farmers. Alarmingly, Polymyxin (19%) which is listed as a 'Reserve' by the WHO is the second most abundantly used antibiotics (Fig 3).

A complete disregard to AMR stewardship combined with lack of knowledge among poultry farmers can have devastating impact on propagation of AMR. Widespread use of strong antibiotics and indifference to implementing withdrawal period not only aggravates the AMR situation in Nepal but also pose serious food-safety risks. Withdrawal period is a withhold time prior to market distribution of poultry products after antibiotic usage in production; this is to ensure antibiotics have been degraded sufficiently and rendered inactive [44]. A survey conducted in Kathmandu of more than 200 farmers revealed only few poultry farmers knew about withdrawal periods [7] or were aware of the importance of adhering to withdrawal periods after antibiotic use to prevent the development of AMR [16, 17]. All of these indicate the need of strict monitoring and control of antibiotic use by concerned government agencies.

Farmers in this study mentioned consulting either veterinarians or veterinary technicians regarding antibiotics usage (Table 4), however, either they did not fully comprehend the gravity of the looming AMR threat, or they disregarded the information they received. Majority of the farmers (88%) claimed to have received trusted information on AMR from various experts, but almost half of the farmers were unclear about AMR and majority (80%) of them were not willing to observe withdrawal period for their products.

In order to enhance biosecurity and reduce the risk of disease outbreaks in the poultry industry, it is necessary to raise awareness among poultry farmers about the usage of antibiotics along with significance of biosecurity measures [17, 45]. This can be done through extensive practical training amongst network of poultry farmers. Proper biosecurity measures, such as the frequent disinfection of farm premises and equipment and the provision of protective clothing and footwear for workers can greatly help in prevention and containment of farm borne diseases. Additionally, reducing the use of antibiotics in poultry production and promoting alternative methods of disease prevention, such as immunization, use of probiotics and immune modulators can play a crucial role in improving poultry health and reducing disease risks.

## Strengths and limitations

Overall, this pilot study provides a snapshot of critical issues in the poultry industry in Nepal pertaining to poor biosafety, biosecurity, and antibiotic usage that have direct implications for public health, food safety, and AMR concerns. Incorporations of multiple pathogens also provides an insight into disease transmission dynamics in small to medium-sized farms in Nepal.

However, there are few limitations of the study. The study's sample size of sampling 30 chickens per farm may limit the generalizability of the findings to the broader poultry industry in different geographical regions. Likewise, the pathogens screened from the farms were based on Biovac's experience with commercial diagnostics analyzing samples from poultry farms around the country. The pathogens could have been selected based on diseases prevalent in the sampling region. We recommend conducting a larger study with a more comprehensive sample size and in additional locations to ensure generalizability of the findings. Assessment of economic impact of the diseases would also provide insights into farmers' usage of stronger or un-prescribed medications. Additionally, bacteria like *E. coli*, *Salmonella*, *Campylobacter*, *Enterococcus spp*., should be included in future studies as they would serve as better indicators of antibiotics usage and AMR issues in poultry farms and laboratory analyses should go a step beyond PCR to sequence the pathogens to provide a deeper understanding of the potential spread of AMR bacteria.

## Supporting information

**S1 Fig. (Mg):** *Mycoplasma gallisepticum* **(Mg) detected in poultry farms of Nuwakot District.** The four farms were numbered from N1 to N4. Each sample represents pooled oral and cloacal samples. The gel was run with ladder in the first well and positive and negative controls in the last two well respectively. **(IBD): Infectious Bursal Disease (IBD) detected in poultry farms of Nuwakot District.** The four farms were numbered from N1 to N4. Each sample represents pooled oral and cloacal samples. The gel was run with ladder in the first well and positive and negative controls in the last two well respectively.
(ZIP)

**S2 Fig. (IAV): Influenza A Virus (IAV) detected in poultry farms of Ramechhap District.** The four farms were numbered from R1 to R4. Each sample represents pooled oral and cloacal samples. The gel was run with ladder in the first well and positive and negative controls in the last two well respectively.
(TIF)

**S3 Fig. (Mg):** *Mycoplasma gallisepticum* **(Mg) detected in poultry farms of Sindhupalchowk District.** The four farms were numbered from S1 to S4. Each sample represents pooled oral and cloacal samples. The gel was run with ladder in the first well and positive and negative controls in the last two well respectively.
(TIF)

**S4 Fig. (Mg):** *Mycoplasma gallisepticum* **(Mg) detected in poultry farms of Ramechhap District.** The four farms were numbered from R1 to R4. Each sample represents pooled oral and cloacal samples. The gel was run with ladder in the first well and positive and negative controls in the last two well respectively. **(Ms):** *Mycoplasma synoviae* **(Ms) detected in poultry farms of Ramechhap District.** The four farms were numbered from R1 to R4. Each sample represents pooled oral and cloacal samples. The gel was run with ladder in the first well and positive and negative controls in the last two well respectively.
(ZIP)

**S5 Fig. (Mg):** *Mycoplasma gallisepticum* **(Mg) detected in poultry farms of Kavre District.** The four farms were numbered from K1 to K4. Each sample represents pooled oral and cloacal samples. The gel was run with ladder in the first well and positive and negative controls in the last two well respectively.
(TIF)

**S1 Table. Biosafety and biosecurity checklist used to assess the farms.**
(DOCX)

**S2 Table. Antibiotic stewardship survey used to assess farm owners' knowledge on antibiotics and their usage.**
(DOCX)

## Acknowledgments

We would like to thank Dr. Patrick Gan, Mr. Jay Pal Shrestha, and Ms. Sulakchana Rai of ESTH office for all their support. We would also like to show our gratitude to the Department of Livestock Services. Finally, we would also like to thank Prajwol Manandhar for his invaluable work in supervising primer designs and QGIS work along with, laboratory associates of BIOVAC Nepal and CMDN for their help in analyzing the samples.

## Author Contributions

**Conceptualization:** Ajit Poudel, Rajindra Napit, Dibesh B. Karmacharya.

**Data curation:** Shreeya Sharma.

**Formal analysis:** Ajit Poudel, Shova Bhandari.

**Funding acquisition:** Dibesh B. Karmacharya.

**Investigation:** Ajit Poudel, Shreeya Sharma, Dhiraj Puri.

**Methodology:** Ajit Poudel, Kavya Dhital, Shova Bhandari, Pragun Gopal Rajbhandari, Rajindra Napit.

**Project administration:** Ajit Poudel.

**Software:** Ajit Poudel.

**Supervision:** Ajit Poudel, Dibesh B. Karmacharya.

**Validation:** Ajit Poudel.

**Writing – original draft:** Ajit Poudel, Dibesh B. Karmacharya.

**Writing – review & editing:** Ajit Poudel, Dibesh B. Karmacharya.

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
