## [Decision Letter · Decision Letter 0]

29 May 2023

PONE-D-23-10881Poor biosafety and biosecurity practices and haphazard antibiotics usage in poultry farms in Nepal hindering antimicrobial stewardship.PLOS ONE

Dear Dr. Karmacharya,

Thank you for submitting your manuscript to PLOS ONE. After careful consideration, we feel that it has merit but does not fully meet PLOS ONE’s publication criteria as it currently stands. Therefore, we invite you to submit a revised version of the manuscript that addresses the points raised during the review process.

We look forward to receiving your revised manuscript.

Kind regards,

Anselme Shyaka, Ph.D

Academic Editor

PLOS ONE

Journal Requirements:

a) The name of the colleague or the details of the professional service that edited your manuscript.

b) A copy of your manuscript showing your changes by either highlighting them or using track changes (uploaded as a *supporting information* file).

c) A clean copy of the edited manuscript (uploaded as the new *manuscript* file)”.

"This study was made possible by the funding support from the Regional Environment, Science, Technology and Health (ESTH) Office for South Asia- the US State Department.."

"Author : DBK

Grant Number: SNP40021GR3038

Funder: Regional Environment, Science, Technology and Health (ESTH) Office for South Asia- the US State Department

Funder Website: https://id.usembassy.gov/embassy-consulates/jakarta/sections-offices/environment-science-technology-and-health-office/

NO - the funders did not play any role in the study design, data collection and analysis, decision to publish, or preparation of the manuscript"

6. Thank you for stating the following in your Competing Interests section: "The authors do not have any competing interest"

7. We note that you have stated that you will provide repository information for your data at acceptance. Should your manuscript be accepted for publication, we will hold it until you provide the relevant accession numbers or DOIs necessary to access your data. If you wish to make changes to your Data Availability statement, please describe these changes in your cover letter and we will update your Data Availability statement to reflect the information you provide.

8. Please upload a copy of Figure 4 to 6, to which you refer in your text on page 33 (in PDF format). If the figure is no longer to be included as part of the submission please remove all reference to it within the text.

9. We note that Figure 1 in your submission contain map/satellite images which may be copyrighted. All PLOS content is published under the Creative Commons Attribution License (CC BY 4.0), which means that the manuscript, images, and Supporting Information files will be freely available online, and any third party is permitted to access, download, copy, distribute, and use these materials in any way, even commercially, with proper attribution. For these reasons, we cannot publish previously copyrighted maps or satellite images created using proprietary data, such as Google software (Google Maps, Street View, and Earth). For more information, see our copyright guidelines: http://journals.plos.org/plosone/s/licenses-and-copyright.

(1) You may seek permission from the original copyright holder of Figure 1 to publish the content specifically under the CC BY 4.0 license.  

10. PLOS ONE now requires that authors provide the original uncropped and unadjusted images underlying all blot or gel results reported in a submission’s figures or Supporting Information files. This policy and the journal’s other requirements for blot/gel reporting and figure preparation are described in detail at https://journals.plos.org/plosone/s/figures#loc-blot-and-gel-reporting-requirements and https://journals.plos.org/plosone/s/figures#loc-preparing-figures-from-image-files. When you submit your revised manuscript, please ensure that your figures adhere fully to these guidelines and provide the original underlying images for all blot or gel data reported in your submission. See the following link for instructions on providing the original image data: https://journals.plos.org/plosone/s/figures#loc-original-images-for-blots-and-gels. 

11. Please include a copy of Table 4 to 6 which you refer to in your text on page 34 (in PDF format).

12. Please include captions for your Supporting Information files at the end of your manuscript, and update any in-text citations to match accordingly. Please see our Supporting Information guidelines for more information: http://journals.plos.org/plosone/s/supporting-information.

**Additional Editor Comments:**

Please carefully read the reviewers’ comments and submit a point-by-point reply to each raised question/concern.

In addition to the comments, please make sure that you address the concern regarding the following:

- The ethics statement: Since there have been interactions between the investigators and the participant, this research qualifies as “human subject research.” Has the consent form been obtained before collecting the data? Has the participation been voluntary, and were the participants informed about the risk and benefits of participating in the research?

- Availability of Data: As one reviewer mentioned, and as per PLOS ONE data policy, you must submit the values behind the proportions reported for each BBCM category. Alternatively, you may want to use a public data repository or clarify a contact person if authors do not have the right to share the data.

- Grammar and typographical errors: Please recheck the manuscript before re-submission. I encourage you to use language-editing and copyediting services to ensure meeting PLOS ONE publication criterion for language standards.

Reviewers' comments:

Reviewer's Responses to Questions

**Comments to the Author**

1. Is the manuscript technically sound, and do the data support the conclusions?

Reviewer #1: Yes

Reviewer #2: Partly

2. Has the statistical analysis been performed appropriately and rigorously? 

Reviewer #1: N/A

Reviewer #2: No

3. Have the authors made all data underlying the findings in their manuscript fully available?

Reviewer #1: No

Reviewer #2: Yes

4. Is the manuscript presented in an intelligible fashion and written in standard English?

Reviewer #1: No

Reviewer #2: No

5. Review Comments to the Author

Reviewer #1: In general, the manuscript is written in in standard English but there are several grammar and typographical errors. Those identified together with all the other comments are shown in the attached Word document.

Reviewer #2: Reviewer’ comments

The authors investigated Poor biosafety and biosecurity practices and haphazard antibiotics usage in poultry farms in Nepal hindering antimicrobial stewardship. The findings may be published after major revision.

Please find below my specific comments:

Abstract

• This section lacks conclusion

Methods

• There is no mention about the study design – please consider including such a subsection.

• Data analysis is underdeveloped (only descriptive statistics was computed). I suggest performing inferential statistics, for example, qui-square tests, relative risks.

• How did you score the Biosafety and Biosecurity Compliance Matrix (BBCM) and categorise it into 3 needs to be detailed further. Please elaborate more on that. Did you check any scientific reference?

• Line 92 – Ambiguity here. The authors stressed they did not collect biological samples, but they took oral and cloacal swabs from study birds. Please clarify.

• Line 116 –117: Did the pooling follow any existing protocol that can be referenced?

• Line 209 – I suggest changing the title to perceptions of AMR among farmers. Indeed, most of the questions were yes and no and with such questions, it would be difficult to assess the knowledge.

• Lines 213-215: Penicillins are also ß-lactams. Again, you mixed names of individual antibiotics (colistin) with those of classes of antibiotics. I think it would be better to homogenize. For example, polymyxins (i.e. colistin),

Discussion

• Consider revisiting the section – some information is not fit for the section rather it is suitable for the introduction or for deleting.

• Lines 222-233 – would be suitable for the introduction section

• Is there any study limitations to highlight?

• There is no section dedicated to conclusions

References

• Consider crosschecking for completeness – for example is there a link to the 1st reference?

Fig. 3:

• Typos - macrolide and pleuromutilin.

Supplementary Table 2: Antibiotic stewardship survey used to assess farm owners’ knowledge on antibiotics and their usage.

• Consider replacing “knowledge” with perceptions.

Supplementary Table 1: Biosafety and biosecurity checklist used to assess the farms

• Water supply – How did the authors assess drinking water standards (i.e., colony count ≤ 1000, E. coli – Nil, Coliforms ≤ 100)? Did they perform bacteriological culturing to assess the level of contamination?

6. PLOS authors have the option to publish the peer review history of their article (what does this mean?). If published, this will include your full peer review and any attached files.

Reviewer #1: No

Reviewer #2: No

---

## [Author Response · Author response to Decision Letter 0]

10 Sep 2023

PONE-D-23-10881 – Review 

The authors did a good job explaining the issues with biosafety and biosecurity measures in Nepal and how that leads to increased AMR and risk for spread of infectious diseases. In general, the study is important as it highlights some of the issues of global health importance like antimicrobial resistance and inadequate biosafety and biosecurity measures. I have included here comments that the authors can use to improve their manuscript.

Title: This is not a major issue, but the title can be rephrased for clarity. 

One example: Antimicrobial stewardship hindered by inadequate biosecurity and biosafety practices, and inappropriate antibiotics usage in poultry farms of Nepal.

ANS – Title has been rephrased.

INTRODUCTION:

The authors do not clearly show the identified gaps they wanted to address through this study. I think that the last paragraph could include a few more details to explicitly indicate: 

• The gaps in knowledge in relation to biosafety and biosecurity.

• Specific research questions they sought to address in this study.

ANS – Introduction section has been reworked.

METHODOLOGY:

It is understood that the authors wanted to conduct "a comprehensive risk assessment and status evaluation of biosafety, biosecurity and AMR stewardship". The study design should be revised to indicate how each component of the methodology is addressing the stated research question/specific objectives. The following points should be made clear in their study design:

Choice of farms:

In the introduction the authors show that contributions of commercial and backyard are in comparable proportions (56% vs 46%) in Nepal. It would be important to investigate both types of operations. The authors should indicate if this was considered in their study design (or why not that was not the case). 

ANS – addressed in lines 118-120 under ‘Methodology’.

How many farms are in each district and how were the four farms selected?

ANS – due to budgetary constraints and to ensure uniformity among the sampling sites, four small-medium farms were selected in each district. This is a limitation of the study.

Were these broiler or layer farms?

ANS – Two broiler and two layer farms were selected in each district. This details has been added in methodology as well. A table has been added with farm details (Table 1).

Choice of organisms to investigate:

What research question did the authors want to address by collecting samples and detecting pathogens by PCR? Although not explicitly stated, it can be understood that this is part of "status evaluation of biosafety, biosecurity and AMR stewardship". The authors chose only viral pathogens and two Mycoplasma species. Some bacteria like Salmonella, E. coli, Avibacterium paragallinarum, ... could have been a good indicator or AMR status since antibiotic usage and AMR are a major part of the study. There might be reasons why these were not included. These should be stated and indicated as one of the study limitations.

ANS – Indeed, and it is a limitation of the study but the viral and mycoplasma pathogens were selected based on Biovac Nepal’s experience with commercial diagnostics in Nepal. The pathogens selected were detected the most in commercial farms that Biovac had screened for its commercial activities. There is certainly a selection bias and is noted as a limitation in conclusion of the Discussion.

Additional comments

Line 114: Were they any criteria for selection of the 30 birds for sampling?

ANS – We utilized the practicality of the central limit theorem and selected 30 birds for sampling. It has been mentioned in methodology [lines 123-126].

Lines 116-117: The pooling process is not clear. Was the pooling based on any criteria like age, sex, ...? Showing total numbers before and after pooling will help to understand the pooling process. 

ANS – As we were screening for infectious diseases, we expected the 30 chicken samples from each farm to be relatively homogenous in terms of exposure or infection status. Form each farm, pooling the oral and cloacal swabs would ensure representative data while reducing the number of tests. The objective of the study was to identify infections in the farms as a whole and not determine prevalence of the disease. Thus the samples were pooled together.

Line 126: The authors indicate that they designed their own primers instead of using published ones. This could be warranted if they had reasons to believe that the organisms in the study area may be different from those used in published literature. In that case, they need to show the source of the genome/gene sequences that were used to design the primers and if possible/applicable, indicate the sequence accession numbers in in the public databases.

ANS – A column has been added to the Table 2 with the GenBank accession numbers or the citations used for the reference sequences.

Lines 147 -148: Did the authors need to use a subtitle if there is only one section?

ANS – the subtitle has been removed.

These two statements seem to contradict each other. The authors should clarify that:

Lines 171-172: Only two farms implemented one out of four record keeping practices.

Lines 178-179: Similarly, record keeping was implemented in all but four farms.

ANS – Line 196 - ‘Only two farms implemented one out of four record keeping practices’ has been changed to ‘Only two farms implemented one out of four rodent control practices.’

Discussion: It would be good if the authors can mention a few strengths AND/OR limitations of the study and what gaps still need to be addressed by research.

ANS – this has been added in Discussion

Grammar and typographical errors

There are several grammar and typographical errors. Below are a few that I picked but the authors should make a thorough search of those. An electronic grammar checker can be helpful here.

ANS – the grammar and typographical errors have been fixed

Line 24: Threatening instead of threating.

Line 27: Questionnaires in plural.

Line 33: Scores instead of scored; For personnel instead of by personal ....

Lines 33, 175 & 182: Should it be personnel instead of personal for consistency with lines 28, 153, 156.

/

Lines 34 & 192: Simplify for clarity: why not "At least one of the screened pathogens was ..."

Line 35: pathogen instead of disease.

Line 48: Chickens in plural?; with only a small number.

line 49: Almost half of the poultry production (46%) comes from....

Line 58: Write HPAI in full for the first appearance of the acronym.

Line 64: Poultry productions ....

Line 73: Capitalize and italicize Salmonella.

Line 99: The sampling and survey were ....

Line 184: Use a simple title. Something like "Biosafety and Biosecurity risk assessment results"

Lines 186 & 187: I would suggest the authors to find a different way to highlight the cells that indicated by color-codes or consult with the editorial team for advice as colors may not be seen in tables. 

ANS – the colors have been removed and the text is written in Bold.

Lines 212-215: These are antibiotic families not individual antibiotics. For example, on line 213, I believe that they meant "Tetracyclines were the most used (36%) antibiotics" OR "Tetracycline was the most used (36%) family of antibiotics."

ANS – fixed in manuscript [lines – 239-240]

Figure 1: the figure does not clearly show the geographical relationship between the city of …. and the study sites. The city boundaries would be a good addition. If necessary, a plain map could be used instead of the terrain map. Does the scale bar apply to the top or bottom map?

ANS – An updated map has been uploaded

---

## [Decision Letter · Decision Letter 1]

2 Oct 2023

PONE-D-23-10881R1Antimicrobial stewardship hindered by inadequate biosecurity and biosafety practices, and inappropriate antibiotics usage in poultry farms of Nepal – A pilot study.PLOS ONE

Dear Dr. Karmacharya,

Thank you for submitting your manuscript to PLOS ONE. After careful consideration, we feel that it has merit but does not fully meet PLOS ONE’s publication criteria as it currently stands. Therefore, we invite you to submit a revised version of the manuscript that addresses the points raised during the review process.

We look forward to receiving your revised manuscript.

Kind regards,

Md. Tanvir Rahman, DVM, MSc, PhD

Academic Editor

PLOS ONE

Journal Requirements:

Reviewers' comments:

Reviewer's Responses to Questions

**Comments to the Author**

1. If the authors have adequately addressed your comments raised in a previous round of review and you feel that this manuscript is now acceptable for publication, you may indicate that here to bypass the “Comments to the Author” section, enter your conflict of interest statement in the “Confidential to Editor” section, and submit your "Accept" recommendation.

Reviewer #2: All comments have been addressed

Reviewer #3: (No Response)

2. Is the manuscript technically sound, and do the data support the conclusions?

Reviewer #2: Yes

Reviewer #3: Partly

3. Has the statistical analysis been performed appropriately and rigorously? 

Reviewer #2: (No Response)

Reviewer #3: N/A

4. Have the authors made all data underlying the findings in their manuscript fully available?

Reviewer #2: Yes

Reviewer #3: Yes

5. Is the manuscript presented in an intelligible fashion and written in standard English?

Reviewer #2: Yes

Reviewer #3: Yes

6. Review Comments to the Author

Reviewer #2: The authors addressed the comments made previously, but the categorization of antibiotics still needs to be revised. For example, β lactams include both penicillins and cephalosporins. Again colistin is individual polymyxin. So, line 240-243 can be revised as follows : The farmers were also inquired about various types and proportion of antibiotics they used (Figure 3). Tetracyclines were the most used (36%) antibiotic class, followed by polymyxins (colistin) (14%), quinolones (12%), aminoglycosides (12%) and macrolides (9%). Penicillins (3%) and cephalosporins (2%) were the least used antibiotic classes.

For clarification - https://www.woah.org/app/uploads/2021/06/a-oie-list-antimicrobials-june2021.pdf.

Line 283 : …… polymyxin (colistin)……

Fig. 3

Consider revising classes of the antibiotics - Tetracycline, polymyxin (colistin), quinolone, aminoglycoside, macrolide, sulfonamide, pleuromutilin, penicillin, cephalosporin.

Reviewer #3: The manuscript entitled “Antimicrobial stewardship hindered by inadequate biosecurity and biosafety practices, and inappropriate antibiotics usage in poultry farms of Nepal – A pilot study” describes the biosecurity gaps and antibiotic uses in poultry farms. The Authors identified poultry bacteria and viruses from those farms as well. Actually, they tried to show a relationship between poor biosecurity and antimicrobial use in poultry farms around Kathmandu City, Nepal.

The manuscript has importance in the field of biosecurity and AMR along with antimicrobial stewardship research in poultry farms. However, the manuscript requires some significant corrections before publication.

The title of the manuscript requires a change. The Authors used biosafety in the title and throughout the manuscript. According to WHO, Biosafety is the safe working practices associated with handling of biological materials. This is related to laboratories or any area where biological materials particularly, pathogens are handled. Therefore, to me, biosafety is not associated with poultry farms. I suggest erasing the word biosafety from the title as well as the whole manuscript.

General comment: Minor English collections are required, particularly, grammar and spelling checks. The table and figure should be self-explanatory. Species should be italicized. Please check the references are appropriately selected. Find the other corrections and clarifications below.

Line No. 22-24: Please check the sentence, particularly this part, “can complicate the spread of poultry diseases…”. Maybe it will be, “can enhance the spread of poultry diseases….”

Line no. 70-71: Please mention, for which animal feed unprocessed poultry waste is being used and contributes to the spread of AI. Please check the reference (13) as well for this statement in line no. 70-71.

Line no. 73: Please check the format of writing Escherichia coli and make the first letter of salmonella as capital.

Line no. 86-87: The Authors have justified the cause of the sampling area around the city of Kathmandu. They used a map showing the sample collection area (four districts, including the farms in the districts) for a better understanding of how the city is surrounded by sample collection areas. I suggest adding the Figure 1 in the methods sections.

Line no. 106-107: Marek’s disease virus detection from feather follicle epithelial cells is best. There are some other important samples like chicken dust, bedding material, etc. Why the Authors screened Marek’s disease virus from cloacal and oropharyngeal swabs?

Please mention the age of every farm selected for sample collections.

Line no. 151-152: What is the basis of the categorization as high, medium, and low score? Any references?

The methodology for AMR stewardship is missing. Please add this part to the methodology.

Figure 3: Please check the spelling of macrolide. Did the Authors use the class name of antibiotics (Beta-lactams, quinolone, macrolide, etc. in the questionnaire? If yes, colistin should be polymyxin. However, the question is, why did the authors use the class name? Generally, Farmers may not be able to understand an antibiotic class. In that case, how did the Authors maintain this interview?

Table 2: What is the meaning of “n”? Be sure that every table is self-explanatory.

7. PLOS authors have the option to publish the peer review history of their article (what does this mean?). If published, this will include your full peer review and any attached files.

Reviewer #2: No

Reviewer #3: **Yes: **Md. Abdus Sobur

---

## [Author Response · Author response to Decision Letter 1]

6 Nov 2023

PONE-D-23-10881R1

Reviewers' comments

Reviewer #2: The authors addressed the comments made previously, but the categorization of antibiotics still needs to be revised. For example, β lactams include both penicillins and cephalosporins. Again colistin is individual polymyxin. So, line 240-243 can be revised as follows : The farmers were also inquired about various types and proportion of antibiotics they used (Figure 3). Tetracyclines were the most used (36%) antibiotic class, followed by polymyxins (colistin) (14%), quinolones (12%), aminoglycosides (12%) and macrolides (9%). Penicillins (3%) and cephalosporins (2%) were the least used antibiotic classes.

For clarification - https://www.woah.org/app/uploads/2021/06/a-oie-list-antimicrobials-june2021.pdf.

ANS – Changes have been made in the manuscript. Penicillins and cephalosporins have been combined into beta-lactams. Image has also been changed.

Line 283 : …… polymyxin (colistin)……

ANS – Changes have been made in the manuscript

Fig. 3

Consider revising classes of the antibiotics - Tetracycline, polymyxin (colistin), quinolone, aminoglycoside, macrolide, sulfonamide, pleuromutilin, penicillin, cephalosporin.

ANS – Changes have been made in the manuscript

The title of the manuscript requires a change. The Authors used biosafety in the title and throughout the manuscript. According to WHO, Biosafety is the safe working practices associated with handling of biological materials. This is related to laboratories or any area where biological materials particularly, pathogens are handled. Therefore, to me, biosafety is not associated with poultry farms. I suggest erasing the word biosafety from the title as well as the whole manuscript.

ANS – According to the CDC, biosafety not only refers to handling of biological materials but also containment of potentially hazardous biological agents to reduce the risk of exposure to personnel or the environment. This could be achieved by using combination of protective barriers like using PPE, appropriate equipment, and conducting risk assessments (Meechan, P. J., & Potts, J. (2020). Biosafety in microbiological and biomedical laboratories.). Therefore, we believe usage of the term is appropriate as inadequate practices in poultry farms can lead to exposure of pathogens to either humans or the environment. 

Additionally, the term biosafety is commonly been used in several publications that address similar issues mentioned in this paper. Few are mentioned below:

Hedman, H. D., Vasco, K. A., & Zhang, L. (2020). A review of antimicrobial resistance in poultry farming within low-resource settings. Animals, 10(8), 1264.

Bello, O. G., Abdulrahaman, O. L., Kayode, A. O., Busari, I. Z., & Koloche, I. M. (2022). Awareness of poultry farmers on biosafety practices against infectious diseases in Kano State, Nigeria. Journal of Agricultural Extension, 26(2), 1-10.. 

Sulzbach, A., Ferla, N. J., da Silva, G. L., & Johann, L. (2022). World occurrence and related problems caused by Megninia ginglymura (Mégnin)(Acari: Analgidae) in commercial poultry farms–a review. World's Poultry Science Journal, 78(1), 215-229.

General comment: Minor English collections are required, particularly, grammar and spelling checks. The table and figure should be self-explanatory 

ANS – Changes have been made to captions for Table 1 and 2 to provide better explanations.

Species should be italicized. 

ANS – All species have been italicized

Please check the references are appropriately selected. Find the other corrections and clarifications below.

Line No. 22-24: Please check the sentence, particularly this part, “can complicate the spread of poultry diseases…”. Maybe it will be, “can enhance the spread of poultry diseases….”

ANS – this has been changed in the manuscript

Line no. 70-71: Please mention, for which animal feed unprocessed poultry waste is being used and contributes to the spread of AI. Please check the reference (13) as well for this statement in line no. 70-71.

ANS – The statement has been edited and appropriate citation has been added.

Line no. 73: Please check the format of writing Escherichia coli and make the first letter of salmonella as capital.

ANS – changed in the manuscript

Line no. 86-87: The Authors have justified the cause of the sampling area around the city of Kathmandu. They used a map showing the sample collection area (four districts, including the farms in the districts) for a better understanding of how the city is surrounded by sample collection areas. I suggest adding the Figure 1 in the methods sections.

ANS – Figure 1 is in the Methodology section.

Line no. 106-107: Marek’s disease virus detection from feather follicle epithelial cells is best. There are some other important samples like chicken dust, bedding material, etc. Why the Authors screened Marek’s disease virus from cloacal and oropharyngeal swabs?

ANS – This was done to maintain uniformity in sampling and we have successfully detected Marek’s disease in diagnostic samples from cloacal and oropharyngeal swabs numerous times prior to and after in the poultry sampled in this study. 

Please mention the age of every farm selected for sample collections.

ANS – This has been added in Table 1.

Line no. 151-152: What is the basis of the categorization as high, medium, and low score? Any references?

ANS – No, we created the scoring matrix.

The methodology for AMR stewardship is missing. Please add this part to the methodology.

ANS – We had developed a questionnaire for assessing AMR stewardship practices and it is mentioned in methodology (line – 124)

Figure 3: Please check the spelling of macrolide. Did the Authors use the class name of antibiotics (Beta-lactams, quinolone, macrolide, etc. in the questionnaire? If yes, colistin should be polymyxin. However, the question is, why did the authors use the class name? Generally, Farmers may not be able to understand an antibiotic class. In that case, how did the Authors maintain this interview?

ANS – Spelling of macrolide has been corrected and colistin has been changed to polymyxin (changes have been made to the manuscript as well). 

We asked the names of individual brand of antibiotics used by the farmers and grouped them according to their class as different antibiotics of the same classes were used in different farms. Thus, to avoid confusion, we used class of those antibiotics in the manuscript.

Table 2: What is the meaning of “n”? Be sure that every table is self-explanatory.

ANS – N stands for nucleoprotein – it has been added to the manuscript. Caption of the table has also been edited.

---

## [Editor Report · Decision Letter 2]

26 Dec 2023

Antimicrobial stewardship hindered by inadequate biosecurity and biosafety practices, and inappropriate antibiotics usage in poultry farms of Nepal – A pilot study.

PONE-D-23-10881R2

Dear Dr. Karmacharya,

We’re pleased to inform you that your manuscript has been judged scientifically suitable for publication and will be formally accepted for publication once it meets all outstanding technical requirements.

Kind regards,

Md. Tanvir Rahman, DVM, MSc, PhD

Academic Editor

PLOS ONE

Additional Editor Comments (optional):

Thanks for updating the manuscript as per comments of the reviewers.
---

## [Editor Report · Acceptance letter]

20 Feb 2024

PONE-D-23-10881R2 

PLOS ONE

Dear Dr. Karmacharya, 

I'm pleased to inform you that your manuscript has been deemed suitable for publication in PLOS ONE. Congratulations! Your manuscript is now being handed over to our production team.

Kind regards, 

on behalf of

Professor Md. Tanvir Rahman 

Academic Editor

PLOS ONE